# Optimization of a Pressurized Extraction Process Based on a Ternary Solvent System for the Recovery of Neuroprotective Compounds from *Eucalyptus marginata* Leaves

**DOI:** 10.3390/ijms26010094

**Published:** 2024-12-26

**Authors:** Soumaya Hasni, Hajer Riguene, Jose A. Mendiola, Elena Ibáñez, Lidia Montero, Gloria Domínguez-Rodríguez, Hanene Ghazghazi, Ghayth Rigane, Ridha Ben Salem

**Affiliations:** 1Laboratory of Organic Chemistry LR17ES08, Faculty of Sciences of Sfax, University of Sfax, B.P 1171, Sfax 3038, Tunisia; soumayahasni77@gmail.com (S.H.); hajer.riguene@yahoo.fr (H.R.); gaith.rigane@yahoo.fr (G.R.); ridhabensalem@yahoo.fr (R.B.S.); 2Foodomics Laboratory, Bioactivity and Food Analysis Department, Institute of Food Science Research CIAL (CSIC-UAM), Calle Nicolas Cabrera 9, 28049 Madrid, Spain; j.mendiola@csic.es (J.A.M.); lidia.montero@csic.es (L.M.); gloria.dominguezr@uah.es (G.D.-R.); 3Laboratory of Management and Valorization of Forest Resources, National Research Institute of Rural Engineering, Water and Forestry (INRGREF), Ariana 2080, Tunisia; hanene8116@yahoo.fr; 4Physic and Chemistry Department, Faculty of Sciences and Technology of Sidi Bouzid, University of Kairouan, B.P «380», Sidi Bouzid 9100, Tunisia

**Keywords:** *Eucalyptus marginata*, supercritical fluid extraction, green solvent, acetylcholinesterase inhibition, phenolic compounds, GC-MS analysis, LC-ESI-MS/MS

## Abstract

Green chemistry focuses on reducing the environmental impacts of chemicals through sustainable practices. Traditional methods for extracting bioactive compounds from *Eucalyptus marginata* leaves, such as hydro-distillation and organic solvent extraction, have limitations, including long extraction times, high energy consumption, and potential toxic solvent residues. This study explored the use of supercritical fluid extraction (SFE), pressurized liquid extraction (PLE), and gas-expanded liquid (GXL) processes to improve efficiency and selectivity. These techniques were combined in a single mixture design, where CO_2_ was used in the experiments carried out under SFE, while water and ethanol were used for the PLE and GXL experiments by varying the concentration of the solvents to cover all the extraction possibilities. The neuroprotective activity of the extracts was evaluated by measuring their antioxidant, anti-inflammatory, and acetylcholinesterase inhibition properties. The optimization resulted in a novel GXL extraction with an optimal ternary mixture of 27% CO_2_, 55% ethanol, and 18% water, with a high degree of desirability (R^2^ = 88.59%). Chromatographic analysis carried out by GC-MS and HPLC-ESI-MS/MS identified over 49 metabolites. The designed sustainable extraction process offers a promising approach for producing phenolic-rich plant extracts in industrial applications.

## 1. Introduction

Medicinal and aromatic plants are valuable sources of bioactive compounds, particularly phenolic compounds, which have shown potential in reducing the risks of diseases, like cancer, cardiovascular conditions, and Alzheimer’s disease, among others [1]. These compounds, including flavonoids, stilbenes, and phenolic acids, are known for their potential bioactive properties, such as antioxidant, anti-inflammatory, and neuroprotective properties, positioning them as natural alternatives to synthetic drugs [2]. Among the wide variety of plant species, *Eucalyptus marginata* (Jarrah) stands out for its potential beneficial effects, particularly because of the bioactive compounds present in its leaves [3]. Traditionally used to treat ailments, such as asthma, respiratory inflammation, and rheumatic pain, *E. marginata* has gained scientific attention for its rich phytochemical profile, which includes terpenoids, phenolic acids, flavonoids, and tannins [4,5].

In recent years, sustainable extraction processes have become a key focus in green chemistry, aiming to reduce the environmental impacts associated with traditional techniques. Traditional solvent-based protocols often involve long extraction times, high energy consumption, and the use of toxic organic solvents, which can degrade bioactive compounds and leave harmful residues. To overcome these limitations, innovative techniques, such as supercritical fluid extraction (SFE), pressurized liquid extraction (PLE), and gas-expanded liquid extraction (GXL), have been developed [6]. These techniques offer improved efficiency, reduced solvent use, and minimal degradation of bioactive compounds, making them ideal for extracting thermolabile substances from plants, like *E. marginata* [7].

CO_2_ is a widely used solvent in SFE because of its cost effectiveness, non-toxicity, and ability to operate at low temperatures and pressures [8]. At its critical point (30.9 °C and 73.8 bar), CO_2_ exhibits properties that enable deep matrix penetration. Moreover, the use of CO_2_ allows for easy solvent removal at the end of the process, after the depressurization step, without leaving residues in the extract and making it an environmentally friendly alternative to organic solvents, like hexane or dichloromethane. Other advantages of SFE are that supercritical CO_2_ operates in an oxygen-free environment, which further enhances its utility for extracting sensitive compounds, as well as the short extraction times and minimal use of additional organic solvents that offer distinct advantages over traditional techniques [9].

On the other hand, PLE is an advanced technique that uses liquid solvents at elevated pressures and temperatures above the boiling point and below the critical point of the liquid to enhance the solvent penetration of plant matrices, resulting in higher extraction yields and shorter processing times. PLE allows for greater control over extraction conditions, such as the temperature, solvent-to-matrix ratio, and extraction time, which are critical for optimizing efficiency and maintaining the quality of the extracted compounds. Automated PLE instruments further enhance reproducibility and reduce labor, making the technique widely applicable in food and pharmaceutical industries [10].

GXL represents a more recent innovation in extraction technology, offering highly adjustable thermodynamic properties by combining liquid solvents with gases or supercritical fluids [11]. GXL allows for the fine-tuning of extraction conditions, such as the pressure, temperature, and solvent-to-gas ratio, resulting in enhanced efficiency and improved sustainability. Their ability to expand in volume under pressure and contract with temperature changes provides unique advantages, such as increased mass transfer, which are particularly useful for extracting bioactive compounds from natural products [12]. GXL has gained attention for its environmental benefits, as it reduces the use of toxic solvents and integrates well with analytical techniques used in foodomics and natural product studies [12].

The current study aims to explore the extraction of bioactive compounds from *E. marginata* leaves, considering the use of SFE, PLE, and GXL conditions and their combination to maximize the extractability of all the bioactive compounds’ families present in the leaves. The obtained extract was evaluated for its neuroprotective activity and its chemical composition. These processes offer a sustainable and efficient approach to the extraction of valuable bioactive compounds, contributing to the advancement of green chemistry and the development of natural-product-based therapies.

## 2. Results and Discussion

### 2.1. Yields and Kinetic Extraction Study

A kinetic study was conducted to assess the extraction efficiency of polyphenols from *E. marginata* powder, using supercritical CO_2_ and comparing its behavior with those of other solvents, such as ethanol, water, and the different combinations proposed in the mixture design (see Figure 1a). For this evaluation, 2 g of leaf powder was extracted at a flow rate of 4 mL/min, 50 °C, and 100 bar.

To investigate the total extraction time, samples were sequentially collected every 15 min throughout the extraction process (120 min), see Figure 1b. The in vitro assays of these extracts were subsequently measured in order to evaluate the behavior of the phenolic compound extraction along the process to establish the final extraction time.

Statistical analysis confirmed no significant changes in yield with longer extraction times. Thus, to optimize energy consumption and minimize potential undesirable reactions, 45 min was chosen as the optimal extraction time. The total phenolic compounds did not significantly increase after 45 min of extraction (Figure 1a).

The extraction solvent that provided higher extraction yields was the test performed using 25% CO_2_-50% ethanol–25% water (Table 1). This suggests that expanded-gas (low-density CO_2_) extractions were more effective compared to pure CO_2_. These findings align with previous studies, which demonstrated that water–ethanol mixtures in SFE may enhance the extraction yields of polar bioactive compounds, such as phenolic compounds [13,14]. Table 2 summarizes the extraction yields obtained under identical conditions (50 °C, 100 bar, and 4 mL/min for 2 h) using all the different solvent compositions from the mixture design. For comparison, SFE with pure CO_2_ at 350 bar, 50 °C, and 4 mL/min for 2 h was also performed, and these results are included. Additionally, conventional maceration extractions were conducted using 100% acetone, 100% ethanol, and 100% water (see Section 3 for details).

These results show that ternary mixtures and binary pressurized water–ethanol mixtures resulted in the highest extraction yields, while pure CO_2_ displayed the lowest, even at high pressures (350 bar). The highest yield (54%) was achieved with 25% CO_2_, 50% EtOH, and 25% H_2_O, suggesting that this combination of CO_2_, ethanol, and water as a solvent is optimal for maximizing the extraction yield. Notably, the pressurized extractions yielded significantly higher phenolic compound contents compared to those yielded by traditional maceration (for 24 h at 25 °C).

### 2.2. Phenolic Compound and Flavonoid Quantification

Significant variations in TPC were observed across different extraction conditions, with the highest TPC (660 ± 3 mg GAE/g extract) obtained using the ternary solvent mixture of 25% CO_2_, 50% ethanol, and 25% water. This solvent combination also produced the highest extraction yield, indicating its efficiency in extracting phenolic compounds.

Pure pressurized solvents, such as 100% ethanol (407 ± 6 mg GAE/g extract) and 100% water (306.1 ± 0.5 mg GAE/g extract), resulted in lower TPC values, underscoring the importance of mixed solvents for maximizing phenolic extraction; in this sense, the highest TPC values were recorded for extracts obtained with pressurized hydroethanolic mixtures (50:50 *v/v*) at a concentration of 633 ± 2 mg GAE/g extract. This variability in the TPC between the different extraction conditions is likely because of the presence of a large variety and the biochemical compositions of samples and the polarity of the solvents used. These results align with findings by [15].

When CO_2_ was used as the solvent, pure CO_2_ at 100 or 350 bar (SFE conditions) resulted in low TPC values, not exceeding 42 ± 3 and 51 ± 1 mg GAE/g extract, respectively. In contrast, ternary mixtures of CO_2_, ethanol, and water (GXL) yielded a wider range of TPC values. As reported by Solana and his team, it is clear that the inclusion of water–ethanol mixtures in supercritical CO_2_ significantly improves the extraction of phenolic compounds, probably because of the wide range of polarities found in these compounds. This mixture yielded the highest total phenolic content, at 2.78 mg/g.

The lowest TPC value achieved with the ternary mixture (376 ± 5 mg GAE/g extract) was observed with 25% CO_2_, 25% ethanol, and 50% water, while the highest TPC values (629 ± 1 and 660 ± 3 mg GAE/g extract) were achieved with solvent combinations of 50:25:25 (*v/v*) and 25:50:25 (*v/v*) of CO_2_, ethanol, and water, respectively, which, indeed, were the conditions that provided higher TPC values along the whole experimental design. The central point mixture (33.3% CO_2_, 33.3% ethanol, and 33.3% water) also produced significant TPC values, highlighting that the addition of water and ethanol to CO_2_ enhances phenolic extraction, likely because of the broad polarity range of phenolic compounds.

These results are consistent with those in previous studies [16,17], which demonstrated that although water alone provided the highest extraction yield, ethanol addition increased the phenolic content. This suggests that ethanol’s ability to dissolve a diverse range of compounds can increase selectivity, resulting in higher phenolic extraction rates.

Concerning the maceration extraction, the highest TPC values were obtained with 100% ethanol (386 ± 8 mg GAE/g extract), followed by 100% acetone (334 ± 22 mg GAE/g extract), while 100% water yielded a lower TPC of 203 ± 5 mg GAE/g extract. Comparing these results with those obtained with 25:50:25 (*v/v*) of CO_2_, ethanol, and water in pressurized extraction, the pressurized extraction techniques provided between 3.3 and 2 times higher phenolic compound contents.

A statistically significant correlation was found between phenol and flavonoid levels across different extraction conditions, TFC = 0.4329 TPC − 1.6715; R^2^ = 0.9442. Similar to TPC, the highest TFC (289 ± 7 mg QE/g extract) was obtained with 25% CO_2_, 50% ethanol, and 25% water, indicating that solvent mixtures are more effective than pure solvents for flavonoid extraction. Pure CO_2_ (100%) at both 100 and 350 bar resulted in the lowest TFC values, reinforcing the inefficiency of CO_2_ alone for flavonoid extraction. Notably, extracts with higher phenolic contents also exhibited higher flavonoid contents, supporting the findings of Grichi et al. and Nagappan et al. [18,19].

### 2.3. Biological Activity Evaluation

#### 2.3.1. Antioxidant Activities

Natural antioxidants are critical in neutralizing free radicals, which play a key role in the progression of oxidative stress. *E. marginata* leaf extracts demonstrated strong antioxidant capacity across several assays, including DPPH, ABTS-TEAC, and ORAC, supporting findings from previous studies [20].

In the DPPH assay, IC_50_ values for pressurized extraction ranged from 6.8 ± 0.5 at 50% CO_2_ and 50% ethanol to 80 ± 9 μg/mL at 100% CO_2_ (100 bar) (lower values indicate higher antioxidant capacities), whereas maceration extractions yielded values between 10.3 ± 0.3 at 100% ethanol and 20 ± 2 μg/mL at 100% water. The strongest antioxidant activity was observed in extractions using 50% CO_2_ and 50% ethanol (IC_50_ = 6.8 ± 0.5 μg/mL) and 25% CO_2_, 50% ethanol, and 25% water (IC_50_ = 7.1 ± 0.3 μg/mL), highlighting the higher efficiency of these mixed solvent systems. Similarly, the ethanol–water (50:50 *v*/*v*) mixture at 100 bar pressure showed an IC50 of 9.5 ± 0.2 μg/mL significant antioxidant potential.

These results emphasize the relationship between antioxidant activity and the concentration of active compounds in the extract. These results suggest that mixed solvent systems enhance the radical-scavenging abilities of *E. marginata* extracts, compared to the use of pure solvents.

The ORAC assay further confirmed the strong antioxidant activity of the extracts. This test evaluates how effectively a sample can absorb oxygen radicals, with fluorescence intensity decreasing as oxidation progresses. As seen in Table 2, extracts obtained using GXL, PLE, pure CO_2_ (SFE), and maceration demonstrated strong antioxidant capacities, with IC_50_ values ranging from 3.05 ± 0.8 to 7.9 ± 0.2 μg/mL extract, revealing, once more, that the combination of the CO_2_–ethanol–water (25:50:25, *v/v/v*) mixture was the extract with a higher ORAC activity This consistency across DPPH, ABTS, and ORAC assays indicates the high radical-scavenging potential of the phenolic compounds present in *E. marginata* leaves. The lowest IC_50_ values were recorded for the mixed solvent systems, particularly those involving the ternary solvent mixture of CO_2_, ethanol, and water, further highlighting the enhanced efficiency of these solvent combinations in extracting bioactive compounds with potent antioxidant properties. This finding reinforces previous studies, such as those by Park et al. (2023), which reported that the 30% ethanol extract showed the strongest antioxidant activity against DPPH radicals, while both 30% and 50% ethanol extraction conditions provided the best efficacy for ABTS radical scavenging [21]. Other findings from our study indicate a significantly higher antioxidant potential in *E. marginata* extracts when using 100% EtOH, with an IC_50_ value of 10.3 ± 0.3 μg/mL, compared to 23.204 μg/mL, as reported by Hasni et al. (2021), for extracts obtained with 60% EtOH [2]. This discrepancy suggests that the solvent used in the extraction process plays a crucial role in determining the antioxidant capacity of the extracts [2].

#### 2.3.2. Anticholinergic Activity (AChE)

Acetylcholine, a crucial neurotransmitter in the nervous system, facilitates signal transmission between nerve cells and between nerve cells and muscles. It plays vital roles in muscle contraction, cognition, memory, and the regulation of the autonomic nervous system, considering the multifactorial nature of neurological disorders, such as Alzheimer’s and Parkinson’s diseases [22]. The acetylcholinesterase (AChE) inhibition of the extracts was measured as a bioactivity highly related to these disorders.

The results indicate that all the extracts obtained from *E. marginata* leaves exhibited AChE inhibitory activity. The IC_50_ values ranged from 43 ± 4 to 140.2 ± 0.8 μg/mL extract. The most potent AChE inhibition was observed with CO_2_–ethanol (50:50, *v/v*) (43 ± 4 μg/mL extract). This suggests that this combination is highly effective for acetylcholinesterase inhibition, which is relevant for neuroprotective studies. Among the extracts, those obtained using PLE with 100% ethanol and a 50:50 mixture of ethanol and water also showed high AChE inhibition rates, with IC_50_ values of 66 ± 4 and 75.06 ± 0.02 μg/mL extract, respectively.

For GXL extraction, the best inhibition was observed with CO_2_–EtOH (50:50, *v/v*) and a ternary mixture of CO_2_–ethanol–water (25:50:25, *v/v/v*), yielding IC_50_ values of 43 ± 4 and 45 ± 1 μg/mL extract, respectively. In contrast, compounds extracted using SFE with 100% CO_2_ exhibited the lowest activities, with an IC_50_ value of 140.2 ± 0.8 μg/mL extract. Investigations into other species have shown that particular combinations of solvents can effectively extract bioactive compounds linked to neuroprotective effects.

The findings of the current study suggest a strong correlation between the polyphenol content of the extracts and AChE inhibition, implying that the phenolic compound content in *E. marginata* may be responsible for the observed neuroprotective activity.

For instance, González-Burgos et al. (2018) explored the neuroprotective effects of various phenolic compounds, demonstrating their ability to inhibit AChE activity and reduce oxidative stress in neuronal cells [6]. Similarly, Amessis-Ouchemoukh et al. (2014) reported that extracts from different plant species exhibit varying degrees of AChE inhibition, reinforcing the notion that solvent extraction methods significantly influence the bioactive profiles of these extracts [23].

The finding that maceration with 100% ethanol yielded a high AChE inhibitory activity further corroborates previous reports indicating that ethanol is a particularly effective solvent for extracting phenolic compounds known for their neuroprotective effects. Studies have established that ethanol extracts from various plants consistently exhibit AChE inhibitory properties, making them promising candidates for further investigation in the context of neurodegenerative diseases.

#### 2.3.3. Anti-Inflammatory Activity (LOX Inhibition)

Lipoxygenase (LOX) is an enzyme that catalyzes the oxygenation of polyunsaturated fatty acids to form fatty acid hydroperoxides. LOX plays crucial roles in lipid metabolism and inflammation pathways in both plants and animals [24].

The data in Table 2 show that the anti-inflammatory activities, assessed via LOX inhibition assays, ranged from IC_50_ values of 41 ± 3 μg/mL extract for the CO_2_–ethanol–water (25:50:25, *v/v/v*) mixture and 51.3 ± 0.5 μg/mL extract for the mixture of 50% CO_2_ and 50% ethanol to 147.1 ± 0.6 μg/mL extract for the extract obtained using 100% CO_2_. The strong inhibition of the LOX enzyme highlights the anti-inflammatory potential of *E. marginata* extracts.

In comparison, maceration with 100% ethanol also demonstrated strong anti-inflammatory activity (89 μg/mL extract), followed by the extract obtained using 100% acetone (94 μg/mL extract), while the extract obtained with 100% water presented an IC_50_ value of 112.7 μg/mL extract. These results suggest that ethanol is more effective than acetone and water for the extraction of bioactive compounds with anti-inflammatory properties, during maceration.

Overall, these findings highlight the potential of *Eucalyptus* extracts, particularly those obtained using polar solvents, as natural anti-inflammatory agents through LOX inhibition. The varying degrees of efficacy among different species and extraction methods provide valuable insights for further exploration of *Eucalyptus* as a source of therapeutic agents in managing inflammation-related conditions. Our result is in agreement with those of Hassine et al. (2022), who highlighted the higher inhibition activity of an aqueous extract of *E. citriodora Kino species*, which showed an IC_50_ value of equal to 103.56 mg/mL [25].

### 2.4. Optimization of Extraction Conditions

As previously seen, solvent mixtures, particularly those involving the three solvents CO_2_, ethanol, and water, consistently outperform pure solvents across all the assays, including the extraction yield, TPC, TFC, and bioactivity. The combination of CO_2_–ethanol–water (25:50:25, *v/v/v*) provides the highest values for most of the tested responses; the presence of water in the solvent system significantly enhances the extraction of polar compounds, like phenolics, as demonstrated in similar studies [26,27].

Pure CO_2_, especially at low pressures (100 bar), provided poor performance in terms of the extraction yield, TPC, and bioactivity. This suggests that CO_2_ alone is not suitable for the comprehensive extraction of bioactive compounds from *E. marginata* leaves, which, indeed, was expected because of the low polarity of supercritical CO_2_ and the high polarity of phenolic compounds, which are the main bioactive compounds present in these leaves.

After acquiring all the response variables of the ten experiments, the optimization of the extraction parameters was performed using Statgraphics software (Version 18 x64), focusing on eight key response variables: the extraction yield (Y_e_), total phenolic content (Y_TPC_), total flavonoid content (Y_TFC_), DPPH radical-scavenging activity (Y_DPPH_), ABTS-TEAC antioxidant capacity (Y_ABTS-TEAC_), ORAC capacity (Y_ORAC_), acetylcholinesterase inhibition (Y_AChE_), and lipoxygenase inhibition (Y_LOX_).

A simplex lattice mixture design was employed to optimize the proportions of the three solvent components in the extraction process: CO_2_, ethanol, and water. This type of mixture design allows for the systematic evaluation of how different solvent proportions influence the desired outcomes. By varying the relative concentrations of CO_2_, ethanol, and water, the simplex lattice method effectively models interactions between the components and identifies the most favorable mixture for maximizing bioactive compound extraction.

The optimization demonstrated that the responses were validated with a high desirability index (0.8859). The optimal extraction conditions were identified using GXL extraction with a ternary mixture comprising 27% CO_2_, 55% ethanol, and 18% water. The optimized conditions provided the maximum extraction of biologically active compounds, such as polyphenols and flavonoids, along with enhanced antioxidant, neuroprotective, and anti-inflammatory activities, which is in line with results obtained by Kühn and Temelli [26] and Hassim et al. [28].

Moreover, there was no statistical difference (*p* > 0.05) between the predicted results and those obtained experimentally under the optimal conditions for the extraction yield, total phenolic content (TPC), total flavonoid content (TFC), and various bioactivity measures, highlighting the success of the optimized extraction conditions, as shown in Table 3. An experimental extraction yield of 40% surpassed the predicted 38.24%, suggesting that the conditions favored better extraction. Similarly, the TPC reached 607 mg GAE/g, exceeding the predicted 580.12 mg GAE/g, which indicates the effective extraction of phenolic compounds. The TFC also outperformed expectations, with an experimental value of 274 mg QuE/g compared to the predicted 257.14 mg QuE/g. The DPPH radical-scavenging activity was slightly elevated in the experimental results (3.6 μg/mL) over the predicted 3.04 μg/mL, pointing to increased antioxidant potential. However, the ABTS-TEAC value was lower than anticipated, at 37.94 mmol/g compared to the predicted 43.50 mmol/g, suggesting some variability in the extraction efficiency. The ORAC capacity closely matched predictions, while the acetylcholinesterase inhibition showed stronger experimental results (41 μg/mL) than predicted (46.62 μg/mL), indicating neuroprotective potential. In contrast, the lipoxygenase inhibition was greater than expected, with an experimental value of 49 μg/mL against the predicted 43.50 μg/mL. Overall, these findings suggest that the solvent mixture significantly enhanced the extraction efficiency and bioactivity, although some discrepancies, particularly in antioxidant capacity, merit further exploration. This is the first time this type of experimental design has been applied to the extraction of eucalyptus bioactive extracts; nevertheless, researches have demonstrated that these kinds of ternary mixtures are useful to obtain bioactive extracts [26,27,29] among others. This consistency reinforces the effectiveness of the simplex lattice mixture design in predicting optimal solvent compositions for the extraction of bioactive compounds from *E. marginata* leaves.

### 2.5. Chemical Analysis by GC-MS of Optimal Extract

The volatility profile of the optimal GXL extract (CO_2_–ethanol–water (27:55:18, *v/v/v*)) was analyzed using GC-MS The chromatogram of this extract, as well as the identification of the separated peaks, is shown in Figure 2 and Table 4. Through matching experimental mass spectra with theoretical MS data from databases and verifying the calculated mass’s accuracy, linear retention index, and information available in the existing literature, 28 metabolites were identified.

Based on their chemical structures, these compounds were grouped into several categories, as shown in Table 4. Ketones, such as 5-hexen-2-one (peak 1), are commonly found in *E. marginata*, contributing to its herbal and refreshing aroma. Some of these compounds also possess antiseptic properties, further enhancing the medicinal value of *Eucalyptus* [30]. Terpenoids, including p-cymen-8-ol (peak 2) and cis-beta-terpineol (peak 8), are well known for their antimicrobial and anti-inflammatory effects, making them valuable in skincare formulations [31].

In addition, aldehydes and alcohols, such as 5-hydroxymethylfurfural (peak 3), which is recognized for its antioxidant capabilities and potential anti-cancer properties; cuminaldehyde (peak 4); 4-isopropylbenzyl alcohol (peak 5); and isopinocampheol (peak 9), were detected in significant quantities; these compounds also display antimicrobial and anti-inflammatory activities. These compounds contribute to *Eucalyptus*’s refreshing qualities and may offer various therapeutic benefits. Phenols, such as pyrogallic acid (peak 7) and 2,3-dihydroxy-3-phenylpropanoic acid (peak 10), are recognized for their strong antioxidant and antimicrobial properties, which protect against oxidative stress [32].

Particularly noteworthy is the presence of monoterpenes and sesquiterpenes, including alpha-pinene (peak 6), which can be particularly beneficial in managing conditions like Alzheimer’s disease; ascaridole (peak 13); spathulenol (peak 16); and longipinene epoxide (peak 18). These compounds have demonstrated anti-inflammatory effects and may positively impact neurological function by reducing inflammation in the central nervous system [33].

Table 4 also highlights other compounds, such as carboxylic acids, hydrocarbons, esters, and tocopherols, which contribute to the chemical complexity of *Eucalyptus marginata*. These compounds provide a wide range of benefits, from anti-inflammatory and antioxidant properties to potential neuroprotective effects [34,35].

By integrating these findings with previously cited research on *Eucalyptus* compounds, it becomes clear that the neuroprotective potential of *E. marginata* may be supported by common mechanisms, such as inflammation reduction, modulation of cellular signaling pathways, and protection against oxidative stress. The antioxidant properties of *E. marginata* compounds play a crucial role in protecting neurons from oxidative stress, a major contributor to neurodegeneration. Our study shows that these extracts possess strong radical-scavenging activities, helping to reduce oxidative damage and maintain neuronal integrity. This research highlights the potential of *E. marginata* as a promising source of neuroprotective agents. Future investigations should focus on identifying the specific bioactive compounds and their mechanisms of action.

To summarize, *E. marginata* is highly valued for its aromatic and medicinal properties, which are attributed to a diverse array of volatile chemical compounds, most of them found in the extract optimized in the present work and detected in the GC-MS analysis. These compounds collectively contribute to its therapeutic benefits, particularly in treating respiratory conditions, reducing inflammation, and acting as antimicrobial agents. Moreover, *E. marginata* demonstrates specific neuroprotective qualities, potentially offering protection against neurological disorders [4,36]. Ghazghazi et al. (2019) conducted a comprehensive study on *Eucalyptus marginata*, revealing its antioxidant and antimicrobial properties, which highlight its therapeutic potential in addressing various health issues [37]. In vivo studies could further clarify the therapeutic benefits of *Eucalyptus* extracts in preventing neurodegenerative diseases and promoting cognitive health, in alignment with studies by González-Burgos et al. (2018) and Hassine et al. (2022), which indicated that various *Eucalyptus* species, including *E. globulus*, *E. brevifolia F. Muell*, and *E. stricklandii Maiden*, exhibit neuroprotective activities attributed to their antioxidant properties. This connection underscores the potential of *E. marginata* to contribute to similar protective effects in neurological contexts [6,25].

### 2.6. Chemical Screening of Phenolic Compounds by HPLC-ESI-MS/MS

In this study, the phenolic composition of the *E. marginata* extract was analyzed using HPLC-ESI-MS/MS. The metabolites were annotated using an internal database, which included comparisons of retention times, exact masses, isotopic compositions, UV spectra, and MS/MS data.

Using HPLC-ESI-MS/MS, a total of 16 phenolic compounds were detected in the extract, although only 16 were tentatively identified. The fragmentation of the predominant negative ions in MS/MS mode provided additional insights into the molecular masses and structures of the compounds. Among the identified compounds, phenolic acids and flavonoid derivatives were the most abundant, as detailed in Table 5 and Figure 3.

Chromatographic analysis and mass spectroscopic analysis of the optimal extract from *E. marginata* revealed the presence of several secondary metabolites. Notably, gallic acid was identified (peak 1), with a retention time of 1.8 min, a molecular mass of 170 Da, and a pseudo-molecular ion [M-H]^−^ at *m*/*z* 169. Gallic acid is well-known for its potent antioxidant properties, suggesting that *E. marginata* could offer health benefits, particularly through its ability to neutralize free radicals [38,39].

Other compounds identified include quinic acid (peak 2; Rt = 2.2 min, 192 Da), which has been linked to anti-inflammatory effects and may support immune function [40,41]; protocatechuic acid (peak 4; Rt = 8.6 min, 154 Da), which demonstrates antioxidant and neuroprotective activities, while ellagic acid (peak 6; Rt = 11.7 min, 302 Da) is recognized for its anti-carcinogenic properties [42]. Catechin (peak 7; Rt = 13.8 min, 290 Da) offers cardiovascular benefits through its ability to improve endothelial function. Other identified compounds, such as p-coumaric acid, quercetin-7-*O*-rutinoside, quercetin-3-*O*-rhamnoside, and isorhamnetin, known for their protective effects, also function as antioxidant and anti-inflammatory agents, with potential neuroprotective properties further expanding the complex chemical profile of this *Eucalyptus* species [1]. For example, compound 12 elutes at 21.2 min and corresponds to quercetin-7-*O*-rutinoside, with a pseudo-molecular ion [M-H]^−^ at *m*/*z* 609. Compound 13, characterized by an ion [M-H]^−^ at *m*/*z* 447, is formed by the removal of a glucose unit, identified as quercetin-3-*O*-rhamnoside. The fragmentation of this compound likely involves the cleavage of the glycosidic bond between quercetin and rutinose. This generates the quercetin aglycone, which undergoes rearrangements and modifications, producing derivatives, such as quercetin-3-*O*-rhamnoside, depending on the chemical or enzymatic conditions. The loss of a rhamnose unit results in the formation of isorhamnetin, detected at *m*/*z* 315, as illustrated in Figure 4.

Regarding the peak of compound 14 eluted at 24.6 min, it corresponds to luteolin with a pseudo-molecular ion [M-H]^−^ at *m*/*z* 285. Peak 9 corresponded to kaempherol-3-O-rutinoiside ([M-H]^−^ at *m*/*z* 593); its identification was possible thanks to the presence of a main fragment ion at *m*/*z* 285 ([M-H–rhamnose–hexose]^−^). This process involves the fragmentation of the molecule, which can include the cleavage of glycosidic bonds between kaempferol and rutinose (Figure 5).

These findings emphasize the significance of *E. marginata* not only within natural ecosystems but also as a potential source of bioactive compounds for industrial and medicinal applications. Based on these results, the identified metabolites are considered as abundant in the leaves of *E. marginata* [43,44].

In conclusion, *E. marginata* contains a plethora of phytochemicals with known health benefits. These results validate its traditional use in natural medicine and support continued scientific exploration to develop new therapeutic applications. Because of their antioxidant and anti-inflammatory properties, as well as their ability to modulate cellular pathways, these compounds demonstrate significant potential in protecting and enhancing brain health [45].

## 3. Materials and Methods

### 3.1. Plant Material

The plant species, *Eucalyptus marginata* Donn ex Sm. was identified by Dr. Hanene Ghazghazi using the detailed planting map of the arboretum in Ain Drahem, northwest Tunisia. *E. marginata* leaves were collected in September 2022 at the arboretum in Souiniet, located in the mountainous Kroumirie region. The arboretum is situated at an altitude of 492 m, with geographical coordinates of 35°54′ N latitude and 8°48′ E longitude. The plantation was established in 1959, meaning that the trees were 63 years old at the time of the leaf collection (2022 − 1959 = 63 years).

The collected *E. marginata* leaves were air-dried at room temperature and subsequently ground into powder. The powder was then sieved into different fractions based on the particle size, ranging from 0.125 mm to 2 mm [46]. The resulting material was stored under dark conditions at 4 °C until further use.

### 3.2. Extraction Process and Experimental Design

A mixture design was employed to optimize the extraction of polyphenols, using CO_2_, ethanol, and water as solvents in a way such that the three extraction techniques were tested in a single experimental design [47]. The techniques included supercritical fluid extraction (SFE), pressurized liquid extraction (PLE), and gas-expanded liquid (GXL) processes to improve efficiency and selectivity [48]. Experimental data were analyzed using Statgraphics 18-X64, a statistical and graphical analysis software package [49]. The mixture design aimed to create a predictive model correlating the effects of the temperature, pressure, and solvent (experimental factors) on the extraction yield and bioactivity responses (dependent response variables).

A total of 10 tests were conducted. All the extraction conditions are presented in Table 1. Extractions were carried out at 100 bar and 50 °C, at a flow rate of 4 mL/min for 2 h. Solvents were combined volumetrically, with varying compositions of CO_2_, ethanol, and water (0%, 25%, 33.3%, 50%, and 100% *v*/*v*) to assess the effect of the solvent composition on the extraction efficiency. Moreover, the pressurized extraction was compared with traditional extractions using maceration (for 24 h at 25 °C) using 100% acetone (M1), 100% ethanol (M2), and 100% water (M3).

Another experiment was carried out using 100% CO_2_ at a pressure of 350 bar to assess the performance of pure CO_2_ as a solvent for extracting phenolic compounds. The objective was to determine the extraction efficiency of pure CO_2_ compared to previously tested solvent mixtures, particularly in terms of the yield and total phenolic content. This analysis aimed to explore whether pure CO_2_ could provide benefits, such as higher extraction efficiencies and purities of the extracted compounds.

Each extraction was performed using 2 g of *E. marginata* powder, with a particle size of 0.5 mm. The powder was mixed with 4 g of sand and placed between two layers of glass wool to ensure even distribution during the extraction process. Extractions were performed using a custom-designed compressed-liquid extractor coupled with a high-pressure CO_2_ pump (PU-2080 Plus, Jasco, Hachioji, Japan) and a solvent pump (PU-2080, Jasco Plus, Hachioji, Japan). The equipment and methodology have been previously described in the literature [50].

### 3.3. Phytochemical Screening and Extract Characterization

#### 3.3.1. Total Phenolic Content (TPC)

The total phenolic content was determined using a modified Folin–Ciocalteu method [49]. Briefly, 10 μL of the extract solution (1 mg/mL in 50% ethanol) was mixed with 600 μL of water and 50 μL of undiluted Folin–Ciocalteu reagent. After 1 min, 150 μL of 20% Na_2_CO_3_ (*w/v*) was added, and the volume was adjusted to 1 mL with 190 μL of water. The mixture was incubated in darkness at room temperature for 2 h. Following incubation, 300 μL of each solution was transferred to a 96-well microplate, and absorbance was measured at 760 nm using a Synergy HT microplate reader, Bio-Tek instruments (Winooski, VT, USA). A standard curve was prepared using gallic acid, and TPC results were expressed as gallic acid equivalents (mg GAE/g extract), with an R^2^ value of 0.9962. All the measurements were performed in triplicate.

#### 3.3.2. Total Flavonoid Content (TFC)

The total flavonoid content was determined using quercetin as a standard, with a modified aluminum chloride (AlCl_3_) method [51]. Each well of a microplate was filled with 100 μL of the extract solution (1 mg/mL in 50% ethanol), followed by 140 μL of methanol and 60 μL of 8 mM AlCl_3_. The plate was covered to prevent methanol evaporation and incubated in darkness for 30 min. Absorbance was measured at 425 nm using a spectrophotometer. Quercetin was used for the calibration curve, with results averaged over three replicates (R^2^ = 0.9962).

### 3.4. In Vitro Bioactivity

#### 3.4.1. DPPH Radical-Scavenging Assay

The DPPH free-radical-scavenging capacity was determined using a modified method [48]. Different volumes of the extract solution (0.1 mg/mL), ranging from 10 μL to 100 μL, were added to each well, followed by 150 μL of DPPH solution (6 × 10^−5^ M in methanol). After 30 min of incubation in darkness at room temperature, absorbance was measured at 517 nm using a microplate spectrophotometer. The IC_50_ value, representing 50% radical inhibition, was calculated using linear regression. The extract solvent was used as the blank, and control samples containing only DPPH solution were included.

#### 3.4.2. Antioxidant Capacity of ABTS-TEAC

The ABTS^+•^ cation radical-scavenging activity was determined using a modified method [52,53]. The ABTS^+•^ cation radical was generated by reacting a 7 mM ABTS stock solution with 2.45 mM potassium persulfate in the dark at room temperature for 16 h. The ABTS solution was diluted with phosphate buffer (5 mM, pH 7.4) to achieve an absorbance of 0.700 ± 0.002 at 734 nm at 30 °C. In a 96-well microplate, 100 μL of *E. marginata* extract (0.1–0.5 mg/mL in 50% ethanol) was mixed with 250 μL of ABTS solution and incubated for 45 min in darkness. Trolox was used as the reference standard.

#### 3.4.3. Oxygen Radical Absorbance Capacity (ORAC)

The ORAC assay was conducted following a modified method by Sánchez-Martínez et al. [11]. Extracts and controls (such as ascorbic acid at 0.04 mg/mL) were diluted, and volumes ranging from 100 μL to 0 μL were added to wells, with increasing volumes from 0 to 100 μL of ethanol (10%). Then, 100 μL of phosphate buffer, 100 μL of AAPH (2,2′-azobis(2-methylpropionamidine) dihydrochloride), and 25 μL of fluorescein were added. The reaction was monitored in a spectrophotometer over 1 h at 37 °C, with readings taken every 5 min. The excitation wavelength was set at 485 nm and the emission wavelength at 530 nm.

#### 3.4.4. Acetylcholinesterase Inhibitory Capacity

The AChE inhibition assay was performed using a modified Ellman’s method based on the enzymatic kinetics proposed by Sánchez-Martínez et al. [11]. A solution of acetylcholinesterase (AChE) enzyme was prepared and stabilized in a 150 mM Tris-HCl buffer (pH 8.0). Initially, the Michaelis–Menten constant (K_m_) of the enzyme was determined using acetylthiocholine (AcTh), the substrate, at varying concentrations.

For the inhibition assay, mixtures containing increasing concentrations of the inhibitor/extract (0–100 μL) were prepared, with corresponding decreasing volumes of the solvent (100–0 μL). Each reaction mixture contained 25 μL of the AChE enzyme combined with 100 μL of the Tris-HCl buffer (150 mM, pH 8.0). After 10 min of incubation at room temperature, the reaction was initiated by adding 25 μL of ABD-F (125 μM) in buffer, followed by 50 μL of AcTh (at a concentration based on the calculated K_m_, the substrate concentration required to achieve 50% of the maximum enzymatic velocity).

The mixture was incubated for 15 min at 37 °C, and fluorescence measurements were recorded every minute using a microplate reader set at an excitation wavelength of 389 nm and an emission wavelength of 513 nm. The degree of inhibition (DI) was calculated using the following equation (Equation (1)):DI (%) = V_0_ − V_i_/V_0_
(1)
where V_0_ represents the initial enzymatic velocity in the absence of the inhibitor, and V_i_ is the enzymatic velocity in the presence of the inhibitor. Galantamine was used as a standard inhibitor, and all the measurements were performed in triplicate.

#### 3.4.5. Lipoxygenase (LOX) Inhibitory Capacity

The anti-inflammatory activity was assessed by measuring the inhibition of the lipoxygenase (LOX) enzyme, using a fluorescence-based assay, following the method of Sánchez-Martínez et al. [54]. The Michaelis–Menten constant (K_m_) was first determined to establish the substrate concentration of linoleic acid. For the assay, dilutions were prepared starting from a stock solution of the extract or control (e.g., quercetin at 0.85 mg/mL).

In a 96-well microplate, varying volumes (from 100 to 0 μL) of the extract or inhibitor were combined with corresponding volumes of ethanol (50%) (from 0 to 100 μL). The reaction mixture consisted of 75 μL of the LOX enzyme (0.0208 U/μL) in 150 mM Tris-HCl buffer (pH 9.0), 75 μL of fluorescein, and 100 μL of linoleic acid (K_m_). Fluorescence readings were taken every minute for 15 min at 25 °C, with excitation at 485 nm and emission at 530 nm.

The percentage of the inhibition was calculated using Equation (1), where V_0_ and V*_i_* represent the mean velocities of the LOX in the absence and presence of the inhibitor, respectively. Quercetin was used as a reference inhibitor.

### 3.5. Chemical Characterization of E. marginata Leaf Extracts

#### 3.5.1. GC-MS Analysis

The bioactive compounds extracted via pressurized extraction were analyzed using a gas chromatography–mass spectrometry (GC-MS) system (SHIMADZU QP 2010 Plus, Kyoto, Japan) equipped with an electron ionization (EI) source. Helium was used as the carrier gas at a linear velocity of 32.5 cm/s. Separation was carried out using an Agilent Zorbax DB5-MS + 10 m Duraguard column (30 m × 250 μm × 0.25 μm) with the following oven temperature program: an initial temperature of 45 °C increased to 290 °C at 10 °C/min and finally to 325 °C at 5 °C/min.

The injector temperature was set at 250 °C, with an injection volume of 1 μL in split mode. The mass spectrometer operated in SCAN mode, with an electron impact source at 250 °C and an interfacial temperature of 335 °C. The mass range was set from *m*/*z* 40 to 550, with a scan speed of 1428 amu/s and an event duration of 0.40 s. This method was based on previous protocols by Sánchez-Martínez et al. (2021, 2022) with some modifications [11,54].

Data were analyzed using Shimadzu GC Solution software version 2.53. Compound identification was achieved by comparing acquired mass spectra with reference spectra from the NIST Mass Spectral Library (version 09) and the Wiley Registry of Mass Spectral Data (8th edition).

#### 3.5.2. LC-ESI-MS/MS Analysis

Liquid chromatography coupled to tandem mass spectrometry using an electrospray ionization source (LC-ESI-MS/MS) was used to characterize the phenolic compounds present in the extract. The setup included an Agilent 1100 series (Agilent, Santa Clara, CA, USA) coupled to a Bruker Ion Trap MS Esquire 2000 instrument (Bruker corp., Bremen, Germany).

The analysis was performed on a Phenomenex Gemini NX-C18 column (150 × 2 mm, 3 μm) at 30 °C, at a flow rate of 200 μL/min and an injection volume of 2.5 μL. The mobile phase consisted of (A) water and 0.1% formic acid and (B) acetonitrile and 0.1% formic acid. The gradient program was as follows: 3% B at 0 min, 20% B at 20 min, 75% B at 25 min, 100% B at 35 min, and held at 100% B until 50 min. UV detection was acquired at 254 and 280 nm.

The mass spectrometer operated in negative ion mode, with a nebulizer gas at 25 psi, a dry gas flow rate of 10 L/min, a capillary temperature of 350 °C, and a capillary voltage of 3500 V in an automatic MS/MS fragmentation. This method was adapted from previous studies, with some modifications [55,56].

## 4. Conclusions

This study successfully optimized a pressurized extraction process using GXL with a novel ternary solvent composition (27% CO_2_, 55% ethanol, and 18% water). The GXL method demonstrated superior performance compared to SFE and PLE, achieving high extraction efficiency and producing extracts rich in phenolic compounds and other bioactives. The simplex lattice mixture design effectively identified the optimal conditions for maximizing the recovery of bioactive compounds.

The optimized extraction process facilitated the recovery of neuroprotective compounds from *E. marginata* leaves. The extracts exhibited significant inhibition of acetylcholinesterase (AChE) and lipoxygenase (LOX) enzymes, highlighting their potential in neuroprotection and reductions in inflammatory processes. These findings underscore the efficacy of the simplex lattice experimental design for ternary solvent system optimization to extract compounds with potent biological activities.

In vitro assays confirmed the antioxidant and bioactive properties of the extracts. Assays, such as DPPH, ORAC, and ABTS-TEAC, demonstrated strong antioxidant capabilities, indicating the ability of the extracts to neutralize free radicals and mitigate oxidative stress. Furthermore, the neuroprotective and anti-inflammatory potentials observed in enzymatic inhibition assays provide a foundation for future applications in therapeutic and preventive strategies.

This work establishes the potential of *E. marginata* extracts for use in functional foods, dietary supplements, pharmaceuticals, and cosmetics. The promising neuroprotective, anti-inflammatory, and antioxidant activities encourage further exploration into their mechanisms of action and development for innovative applications in natural medicine and other sciences.

## Figures and Tables

**Figure 1 ijms-26-00094-f001:**
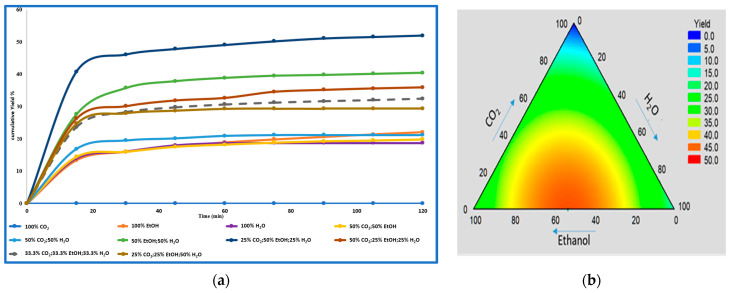
(**a**) Cumulative yield curves for various solvent compositions. (**b**) Total extraction yields represented in a ternary phase diagram.

**Figure 2 ijms-26-00094-f002:**
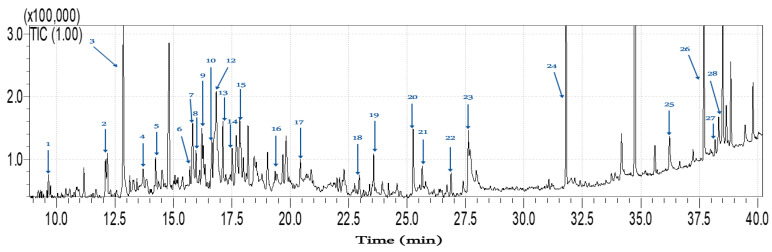
GC-MS chromatogram of the optimal GXL extract from *Eucalyptus marginata* leaves, obtained with CO_2_–ethanol–water (27:55:18, *v*/*v*/*v*) (100 bar, T = 50 °C).

**Figure 3 ijms-26-00094-f003:**
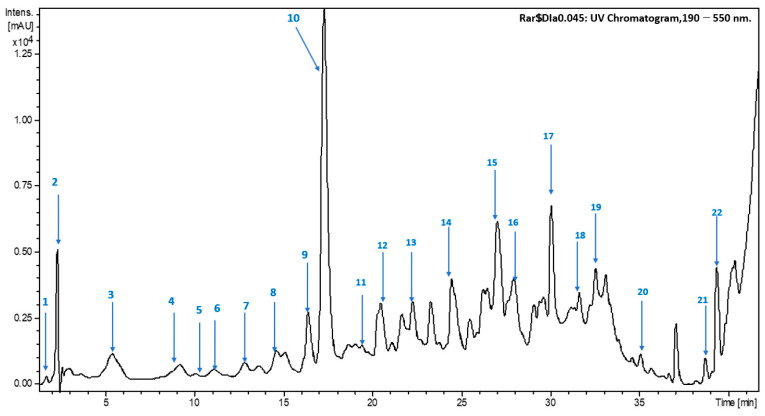
HPLC-ESI-MS chromatogram of the optimal GXL extract from *Eucalyptus marginata* leaves, obtained with CO_2_–ethanol–water (27:55:18, *v*/*v*/*v*) (100 bar, 50 °C).

**Figure 4 ijms-26-00094-f004:**
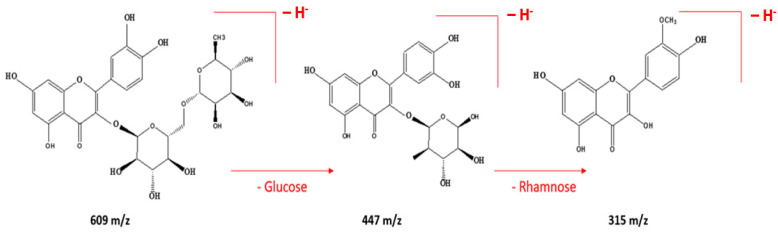
Suggested schematic and chemical structure for the fragmentation of quercetin-7-O-rutinoside (peak 12) in the HPLC-ESI-MS analysis.

**Figure 5 ijms-26-00094-f005:**
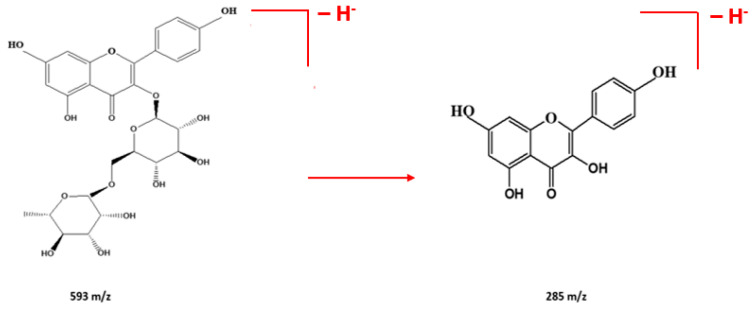
Suggested schematic and chemical structure for the fragmentation of kaempferol-rutinoside in the HPLC-ESI-MS analysis.

**Table 1 ijms-26-00094-t001:** Experimental conditions of extractions performed using the mixture design.

Extraction	CO_2_ (%)	EtOH (%)	H_2_O (%)
1	100	0	0
2	0	100	0
3	0	0	100
4	0	50	50
5	50	50	0
6	0	50	50
7	50	25	25
8	25	50	25
9	25	25	50
10	33.3	33.3	33.3
M1	100% Acetone		
M2	----	100	----
M3	----	----	100

**Table 2 ijms-26-00094-t002:** Extraction yields, total phenolic contents, total flavonoid contents, antioxidant activities (DPPH, ABTS, and ORAC), anti-inflammatory (LOX), and acetylcholinesterase enzyme inhibition (AChE) obtained for all the experimental design points from *E. marginata* leaves, using the pressurized solvent mixtures compared to maceration.

#	Conditions	Yield (%)	TPC mg GAE/g Extract	TFC mg QE/g Extract	DPPH/IC_50_ * (μg/mL Extract)	ABTS (mmol Trolox/g Extract)	ORAC/IC_50_ * (μg/mL Extract)	AChE/IC_50_ * (μg/mL Extract)	LOX/IC_50_ * (μg/mL Extract)
0	100% CO_2_—350 bar	2.2 ± 0.4	51 ± 1	19.3 ± 0.3	56 ± 5	0.91 ± 0.16	6.48 ± 0.09	107 ± 4	140 ± 1
1	100% CO_2_—100 bar	1.1 ± 0.1	42 ± 3	13.9 ± 0.3	80 ± 9	0.66 ± 0.03	7.9 ± 0.2	140.2 ± 0.8	147.1 ± 0.6
2	100% EtOH	22.4 ± 0.3	407 ± 6	194 ± 1	10.9 ± 0.1	5.0 ± 0.1	3.4 ± 0.3	66 ± 4	82.8 ± 0.6
3	100% water	18.6 ± 0.2	306.1 ± 0.5	141.3 ± 0.7	14.6 ± 0.6	3.6 ± 0.1	4.5 ± 0.3	108 ± 4	92.9 ± 0.5
4	50% EtOH; 50% water	42 ± 1	633 ± 2	256 ± 7	9.5 ± 0.2	11 ± 1	3.05 ± 0.08	75.06 ± 0.02	65 ± 2
5	50% CO_2_ 50% EtOH	20.6 ± 0.6	502.2 ± 0.5	241 ± 6	6.8 ± 0.5	13.9 ± 0.3	3.4 ± 0.5	43 ± 4	51.3 ± 0.5
6	50% CO_2_ 50% water	20.8 ± 0.7	466 ± 5	186 ± 3	15 ± 2	8.5 ± 0.2	4.56 ± 0.02	78 ± 1	80 ± 5
7	50% CO_2_; 25% EtOH; 25% water	38 ± 3	629 ± 1	267 ± 7	8.5 ± 0.6	15.9 ± 0.1	3.45 ± 0.04	62 ± 1	44 ± 1
8	25% CO_2_; 50% EtOH; 25% water	54 ± 2	660 ± 3	289 ± 7	7.1 ± 0.3	17 ± 2	2.7 ± 0.1	45 ± 1	41 ± 3
9	25% CO_2_; 25% EtOH; 50% water	32 ± 4	376 ± 5	122 ± 3	12 ± 2	9 ± 2	5.79 ± 0.06	97 ± 4	66 ± 5
10	33.3% CO_2_; 33.3% EtOH; 33.3% water	35 ± 4	440 ± 2	204 ± 4	8.5 ± 0.6	12.5 ± 0.7	4.7 ± 0.3	57 ± 1	46 ± 2
M1	EM—100% Acetone	10 ± 1	334 ± 22	128 ± 4	17.2 ± 0.9	4.4 ± 0.1	6.2 ± 0.5	75 ± 3	94 ± 4
M2	EM—100% EtOH	15.4 ± 0.1	386 ± 8	143 ± 3	10.3 ± 0.3	3 ± 0.9	3.8 ± 0.2	63 ± 1	89 ± 4
M3	EM—100% water	18 ± 3	203 ± 5	86 ± 2	20 ± 2	2.7 ± 0.2	7.5 ± 0.2	85 ± 2	112.7 ± 8.229

* IC_50_ stands for the half-maximal inhibitory concentration; therefore, lower IC_50_ values mean higher activities.

**Table 3 ijms-26-00094-t003:** Predicted optimal and experimental results of extraction using GXL mixture (CO_2_–ethanol–water (27:55:18, *v/v/v*).

Response	Predicted Value	Experimental Value
Yield (%)	38.2371	40 ± 4
TPC mg GAE/g	580.116	607 ± 13
TFC mg QuE/g	257.143	274 ± 4
DPPH/IC_50_ (μg/mL)	3.0407	3.6 ± 0.3
ABTS-TEAC (mmol/g)	43.497	37.94 ± 1.8
ORAC/IC_50_ (μg/mL)	3.2486	3.2 ± 0.8
AChE/IC_50_ (μg/mL)	46.623	41 ± 2
LOX/IC_50_ (μg/mL)	43.497	49 ± 3

**Table 4 ijms-26-00094-t004:** Identified compounds by GC-MS analysis from GXL (CO_2_–ethanol–water (27:55:18, *v*/*v*/*v*)) extracts of *E. marginata* leaves.

Peak	Retention Time (min)	Peak Area	Family	Name	Molecular Ion (*m*/*z*)	Main Fragments (*m*/*z*)
1	9.654	109097	Ketones	5-Hexen-2-one	98	43, 53, 55, 83, 126
2	12.094	158391	Terpenoids	*p*-Cymen-8-ol	150	43, 51, 61, 61, 77, 91, 103, 105, 135
3	12.851	692104	Aldehyde	5-Hydroxymethylfurfural	126	41, 53, 69, 81, 95, 97, 125, 126
4	13.134	75246	Aldehyde	Cuminaldehyde	148	51, 53, 63, 65, 77, 79, 87, 91, 103, 105, 115, 119, 137, 148
5	14.240	134261	Alcohols	4-Isopropylbenzyl alcohol (cuminol)	150	41, 51, 65, 89, 105, 119, 121, 133, 135, 150
6	15.703	78845	Monoterpene	Alpha-pinene	136	41, 55, 57, 65, 70, 81, 84, 91, 93, 95, 109
7	15.818	442417	Phenols	Pyrogallic acid	126	52, 63, 80, 97, 108, 126
8	15.962	181490	Terpenoids	Cis-beta-terpineol	154	43, 51, 55, 65, 69, 77, 83, 93, 105, 121, 139
9	16.215	280557	Monoterpene	Isopinocampheol	154	43, 51, 55, 81, 84, 95, 107
10	16.740	427622	Phenols	2,3-Dihydroxy-3-phenylpropanoic acid	182	51, 58, 63, 65 77, 79, 86, 89, 91, 105, 107
11	16.834	462529	Hydrocarbons	7-Tetradecene	196	41, 53, 55, 67, 69, 71, 111, 126
12	16.900	77975	Carboxylic acid	Spiro[2.2]pentane-1-carboxylic acid, 2-cyclopropyl-2-methyl-	166	43, 53, 55, 65, 67, 77, 79, 105, 107, 121, 125
13	17.116	230775	Monoterpene	Ascaridole	168	41, 55, 57, 69, 79, 81, 95, 97, 107, 119, 125, 135, 139
14	17.688	381086	Sugar	1,6-Anhydro-beta-d-glucopyranose	162	43, 47, 55, 60, 61, 70, 73, 98
15	17.993	97693	Hydrocarbons	1-Dodecyne	166	43, 55, 81, 91, 95
16	19.684	111528	Sesquiterpenes	Spathulenol	220	43, 55, 67, 79, 81, 91, 93, 105, 109, 121, 147
17	20.427	90074	Monoterpene	2-Oxabicyclo[2.2.2]octan-6-ol, 1,3,3-trimethyl-, acetate	212	43, 53, 55, 65, 71, 79, 83, 97, 111, 126, 137, 152
18	22.952	86454	Sesquiterpenes	Longipinene epoxide	220	43, 51, 55, 57, 65, 67, 82, 91, 95, 105, 107, 135
19	23.566	122678	Hydrocarbons	Octadecyne	250	43, 53, 55, 67, 95, 109, 123, 137
20	25.257	280319	Carboxylic acid	Palmitic acid	256	41, 43, 55, 57, 57, 60, 67, 69, 73, 83, 85, 87, 115, 129, 143, 157, 171, 185, 213
21	25.644	81342	Terpenoid	1,2,8,9-Diepoxylimonene	168	43, 44, 53, 55, 71, 79, 95, 107
22	26.878	88571	Monoterpenes	*p*-Menthane-1,8-diol monohydrate	190	43, 51, 55, 59, 65, 72, 91
23	27.618	272678	Ester	Methyl linoleate	294	41, 45, 55, 67, 79, 81, 91, 95, 109, 123
24	31.801	1032336	Alcohols	2,2-Dimethyl-1-hexanol	130	43, 55, 57, 67, 73, 99
25	35.603	152684	Alcohols	trans-2-tetradecenol	212	43, 55, 57, 71, 85, 96, 99, 109, 123, 137
26	36.223	122311	Alcohols	2-Butyl-1-octanol	186	43, 57, 67, 71, 83, 85, 96, 97, 111, 113
27	38.327	233784	Alcohols	Lignoceric alcohol	354	43, 55, 57, 66, 69, 81, 83, 96, 97, 111, 112, 125
28	38.848	371904	Tocopherols	Alpha-tocopherol	430	43, 57, 58, 69, 71, 81, 107, 121, 136, 149, 165, 177, 205

**Table 5 ijms-26-00094-t005:** Analysis of bioactive components from *Eucalyptus marginata* leaf extract by HPLC-ESI-MS.

Peak	Retention Time (min)	Molecular Ion [M-H]^−^ *m*/*z*	Compound
1	1.8	169	Gallic acid
2	2.2	191	Quinic acid
3	7.8	ND	ND
4	8.6	153	Protocatechuic acid
5	10.5	ND	ND
6	11.7	301	Ellagic acid
7	13.8	289	Catechin
8	14.6	207	Sinapaldehyde
9	16.7	593	Kaempherol-3-*O*-rutinoiside
10	17.7	163	p-Coumaric acid
11	19.7	ND	ND
12	21.2	609	Quercetin-7-*O*-rutinoside
13	23.1	447	Quercetin-3-*O*-rhamnoside
14	24.6	285	Luteolin
15	26.6	193	Ferulic acid
16	28.5	271	Naringenin
17	30.4	179	Caffeic acid
18	31.8	315	Isorhamnetin
19	32.4	311	Desmethyleucalyptin
20	35.6	ND	ND
21	38.6	ND	ND
22	40.5	ND	ND

ND: Not determined.

## Data Availability

The data that support the findings of this study are available from the corresponding author upon reasonable request.

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
