# Peer review of "Optimization of a Pressurized Extraction Process Based on a Ternary Solvent System for the Recovery of Neuroprotective Compounds from Eucalyptus marginata Leaves"

_ijms, 2024, doi:10.3390/ijms26010094_

Round 1

Reviewer 1 Report

Comments and Suggestions for Authors

The manuscript is concerned with the Optimization of a pressurized extraction process based on a ternary solvent system for the recovery of neuroprotective compounds from Eucalyptus marginata leaves. It is a good concept and a well-written manuscript with updated references. The conclusions of the work are based on evidence from the work. Generally, the presented manuscript represents a quality with good potential for applicability. However, minor comments are suggested below:

Comments

1- Line 97, please convert “extractability” into “extract ability”.

2- Lines 108-111, please add a reference for this paragraph.

3- Please write “Eucalyptus marginata” for the first time, then after that write “E. marginata”

4- In Table 3, “IC50” should be IC50

5- Line 179, “Sánchez-Martínez et al 2022 [16].” Please follow the reference style throughout the manuscript.

6- Line 254, please convert “Results” into “Results and Discussion.”

Author Response

According to Reviewer: 1

  • Line 97, please convert “extractability” into “extract ability”.

As suggested by the reviewer, the authors have converted extractability by extract ability. Please line 97 page 2.

2- Lines 108-111, please add a reference for this paragraph.

As suggested by the reviewer, the author have added the artcile of Polito et al 2022.

Polito, F.; Kouki, H.; Khedhri, S.; Hamrouni, L.; Mabrouk, Y.; Amri, I.; Nazzaro, F.; Fratianni, F.; De Feo, V. Chemical Composition and Phytotoxic and Antibiofilm Activity of the Essential Oils of Eucalyptus Bicostata, E. Gigantea, E. Intertexta, E. Obliqua, E. Pauciflora and E. Tereticornis. Plants 2022, 11, doi:10.3390/plants11223017.

Please see line 113 page 3.

3- Please write “Eucalyptus marginata” for the first time, then after that write “E. marginata”

As proposed by the reviewer, the authors have changed Eucalyptus marginata by E. marginata in all the manuscript.  

4- In Table 3, “IC50” should be “IC50

As suggested by the reviewer, all the IC50 have been changed to IC50. Please see table 3.

5- Line 179, “Sánchez-Martínez et al 2022 [16].” Please follow the reference style throughout the manuscript.

As indicated by the reviewer, the authors have revised the manuscript to ensure that the reference in the text follows the IJMS style guidelines (“Sánchez-Martínez et al. [16]”) and have updated the corresponding reference in the reference list to the appropriate format as per the journal requirements.

6- Line 254, please convert “Results” into “Results and Discussion.”

As suggested by the reviewer, the authors have changed results by results and discussion.

Please see line 259 page 6.

Reviewer 2 Report

Comments and Suggestions for Authors

Dear Authors,

This study optimized a pressurized extraction process based on a ternary solvent system for the recovery of neuroprotective compounds from Eucalyptus marginata leaves. The topic is very important for plant resource development and utilization. The designed sustainable extraction process offers a promising approach for producing phenolic-rich plant extracts in industrial applications. The manuscript was well written. Before the manuscript was accepted to be published in the journal IJMS, some places need to be revised.

1) This reference “Ghazhazi, H., et al., Revue Roumaine de Chimie, 2019, 64, 1055-1062” should be cited in the paper.

2) p.3, 2.1., Plant Material: Authors should mention the age of eucalyptus trees for leave collection.

3) p.8, 3.2.: Table 1 should be Table 2.

4) It is better to increase one subsection, namely “3.3 Biological activity evaluation”. “3.2.1 Antioxidant activity” was revised as “3.3.1…..”, “3.2.2….” was revised as “3.3.2…”,, and “3.2.3…” should be revised as “3.3.3…”.. The previous “3.3….” should be revised as “3.4….”….

 5) Table 5: Identification of the compounds by UHPLC-ESI-MS is very limited and is not accurate. The detailed separation and structural identification are necessary. Authors should mention it and discuss it in the section 4. Conclusion.

Author Response

According to Reviewer: 2

1) This reference “Ghazhazi, H., et al., Revue Roumaine de Chimie, 2019, 64, 1055- 1062” should be cited in the paper.

As proposed by the reviewer, the authors have cited this reference “Ghazghazi, H.; Essghaier, B.; Riguene, H.; Rigane, G.; El Aloui, M.; Oueslati, M.A.; Ben Salem, R.; Sadfi Zouaoui, N.; Naser, Z.; Laarbi Khouja, M. Phytochemical Analysis, Antioxidant and Antimicrobial Activities of Eucalyptus Essential Oil: A Comparative Study between Eucalyptus Marginata L. And Eucalyptus Paucilora L. Rev. Roum. Chim. 2019, 64, 1055–1062, doi:10.33224/rrch.2019.64.12.05.”

2) p.3, 2.1., Plant Material: Authors should mention the age of eucalyptus trees for leave collection.

As suggested by the reviewer, the authors have added the age of Eucalyptus trees as follow: the plantation was established in 1959, meaning that the trees were 63 years old at the time of collection (2022 - 1959 = 63 years). Please see line 108-110 page 3

3) p.8, 3.2.: Table 1 should be Table 2.

As proposed by the reviewer, the authors have changed table 1 by table 2.

4) It is better to increase one subsection, namely “3.3 Biological activity evaluation”. “3.2.1 Antioxidant activity” was revised as “3.3.1…..”, “3.2.2….” was revised as “3.3.2…”,, and “3.2.3…” should be revised as “3.3.3…”.. The previous “3.3….” should be revised as “3.4….”….

As proposed by the reviewer, the authors have revised the "Biological Activity Evaluation" section as follows:

  • The subsection "3.2.1 Antioxidant activity" has been updated to "3.3.1 Antioxidant activity" as per your suggestion.
  • Similarly, "3.2.2" and "3.2.3" have been revised to "3.3.2" and "3.3.3," respectively, to ensure consistency in the organization of the sections.
  • The previous "3.3" section has been moved and renamed to "3.4" to accommodate the new structure and provide a more coherent flow of information.

Please see line 348- 349- 393 – 437 – 466 page 11 - 12.  

5) Table 5: Identification of the compounds by UHPLC-ESI-MS is very limited and is not accurate. The detailed separation and structural identification are necessary. Authors should mention it and discuss it in the section 4. Conclusion.

We thank the reviewer for this comment. In this work, we did not use UHPLC but HPLC, it was a tipping error in the title of the table that has been corrected in the new version. Regarding the information provided in table 5, as we used a low-resolution mass spectrometer (an ion trap MS) the accuracy of the masses is low (±0.5 amu), so we cannot provide the mass with decimals as it can be done when HRMS is used, instead, we can only achieved a tentative identification. Moreover, to clarify this fact, we have removed the column related to the molecular formula of the tentatively identified compounds, since this information can only be provided when the instrument provides accurate masses. We have also removed this column from the GC-MS table.

Reviewer 3 Report

Comments and Suggestions for Authors

In this work, the authors explored the use of supercritical fluid extraction (SFE), pressurized liquid extraction (PLE), and gas expanded liquid (GXL) processes to improve efficiency and selectivity, combining these techniques in a single mixture design, using CO2 in the experiments carried out under SFE, while water and ethanol were used for the PLE and GXL experiments varying the concentration of the solvents to cover all the extraction possibilities. Besides, the neuroprotective activity of the extracts has been evaluated by measuring their antioxidant, anti-inflammatory, and acetylcholinesterase inhibition properties, achieved important results. However, although the techniques and results described are relevant, I think It's important to improve several aspects related to the clarity, specificity and scientificity of the work. Please consider the following comments and suggestions.

General comment:  Please pay attention when writing the results and discussion. It`s important to write the results in relation to the methodology used. Focus the conclusions on the most relevant aspects of the work.

Line 31: Keywords: Many keywords. Please refer to the IJMS format and instructions for authors.

Line 48: Please include a citation after “among others.”

Line 51: The phrase “among many others” is not correct. Please include a citation or remove

Lines 52-54: Support the information with one or more citations.

Lines 55, 66, 95-96, 104: Write only E. marginata. Check throughout of manuscript 

Line 77: I think it should be: "On the other hand, PLE..."

Line 98: The words “…in terms of…” affects the clarity of the sentence

Line 114: Include the names of the three techniques or add scientific references

Line 115: What experimental design? Please include the name or add scientific references

Line 116: Another statistical program (additional to Statgraphics) was used? If you didn't use another statistical program, why put this phrase (a statistical and graphical analysis software)?. If you used another program, please include the reference.

Lines 179-180; 189, 210; 235: Remove the bold

Line 182: The abbreviation is not necessary if you don´t use it more times throughout the manuscript.

Line 254: Results? Please check the IJMS format (Can be "results and discussion")

Line 266: Figure 1. Please separate each graph with letters "a" and "b" so that the reading of the results is clearer, e.g. Figure 1a and Figure 1b.

Lines 268-269: Placed the complete sentence at the end of the paragraph, after "extraction time..."

Lines 272-274: Please improve the writing. Do not repeat "Figure 1" so many times.

Lines 277-282: Improve the writing. Why do you refer to Table 2 several times?

Lines 290-291: The information is repeated in the methodology. Focus on the results obtained.

Lines 293-294: Please write the results obtained and move the table at the end of section 3.2

Line 295: It´s Table 2, not Table 1

Line 345: Why the statistical correlation not shown in the text, tables or figures?

Line 374: Remove “oxygen radical absorbance capacity”

Line 471: Although the analysis and interpretation of the results are correct, why were bibliographic references not included to strengthen these findings? The absence of citations in the discussion may be questionable. Please check this section

Lines 546-548: If you include details in the legend of table 4, why repeat them here? Simply refer to Table 4.

Table 4. Remove "#" and add "pick"

Lines 592; 658: Use a synonym instead of "in conclusion". There is an exclusive section for conclusions

Table 5: Remove “#”. Includes the meaning of "ND" at the end of the table

Line 666: Please focus the conclusions in a clear and orderly manner on the fundamental aspects of the work: e.g. 1) Optimization of a pressurized extraction process based on a ternary solvent system. 2) Recovery of neuroprotective compounds from Eucalyptus marginata leaves. 3) In vitro assays.

Line 684: Check the phrase “…and materials science”?.

Please check other comments in the attached manuscript

Author Response

According to Reviewer: 3

  • Line 31: Keywords: Many keywords. Please refer to the IJMS format and instructions for authors.

In light of the reviewer's comments, the authors have limited the number of keywords to have more appropriate range (three to ten keywords), as per the instructions for authors. The revised keywords are:

Keywords: Eucalyptus marginata; supercritical fluid extraction; green solvent, acetylcholinesterase inhibition; phenolic compounds; GC-MS analysis; LC-ESI-MS/MS.

We hope this adjustment aligns with the format and meets the requirements of the journal.

  • Line 48: Please include a citation after “among others.”

Following the reviewer's advice, the authors have added the following reference:

Moges, G.W.; Manahelohe, G.M.; Asegie, M.A. Phenolic, Flavonoid Contents, Antioxidant, and Antibacterial Activity of Selected Eucalyptus Species: Review. Biol. Med. Nat. Prod. Chem. 2024, 13, 147–157, doi:10.14421/biomedich.2024.131.147-157. 

Please see line 48.

  • Line 51: The phrase “among many others” is not correct. Please include a citation or remove.

As suggested by the reviewer, the authors have removed “among many others”. Pease see line 51.

  • Lines 52-54: Support the information with one or more citations.

As proposed by the reviewer, the authors have added new citation:

Islam, M.K.; Barbour, E.; Locher, C. Authentication of Jarrah (Eucalyptus Marginata) Honey through Its Nectar Signature and Assessment of Its Typical Physicochemical Characteristics. PeerJ Anal. Chem. 2024, 6, e33, doi:10.7717/peerj-achem.33.

Pease see line 53.

  • Lines 55, 66, 95-96, 104: Write only E. marginata. Check throughout of manuscript.

In accordance with the reviewer's suggestion, we have updated Eucalyptus marginata to E. marginata and have reviewed the entire manuscript for consistency.

  • Line 77: I think it should be: “On the other hand, PLE …”

As suggested by the reviewer, the authors have modified the phrase accordingly.

 Please see line 77.

  • Line 98: The words “…in terms of…” affects the clarity of the sentence.

As indicated by the reviewer, we have revised the sentence to improve its clarity. The phrase “in terms of” has been removed and replaced with more direct language. Please see line 98.

  • Line 114: Include the names of the three techniques or add scientific references.

In response to the reviewer's input, we have added the following phrase: …The techniques included supercritical fluid extraction (SFE), pressurized liquid extraction (PLE), and gas-expanded liquid (GXL) processes to improve efficiency and selectivity… and the reference is as follow:

“Yıldırım, M.; ErÅŸatır, M.; Poyraz, S.; Amangeldinova, M.; Kudrina, N.O.; Terletskaya, N. V. Green Extraction of Plant Materials Using Supercritical CO2: Insights into Methods, Analysis, and Bioactivity. Plants 2024, 13, 1–34, doi:10.3390/plants13162295.” Please see line 115.

  • Line 115: What experimental design? Please include the name or add scientific references

As suggested by the reviewer, the authors have included new reference a follow: “Bali, Y.; Kriker, A.; Abimouloud, Y.; Bouzouaid, S. Improving Thermal Insulation of Fired Earth Bricks with Alfa Plant and Glass Powder Additives: Effects on Thermo-Physical and Mechanical Properties. Case Stud. Therm. Eng. 2024, 53, 103778, doi:10.1016/j.csite.2023.103778”. Please see line 118.

  • Line 116: Another statistical program (additional to Statgraphics) was used? If you didn’t use another statistical program, why put this phrase (a statistical and graphical analysis software)? If you used another program, please include the reference.

In response to this comment, we would like to clarify that Statgraphics 18-X64 was the only statistical and graphical analysis software used for data analysis in this study. The phrase "a statistical and graphical analysis software" was included to emphasize that Statgraphics integrates both statistical modeling and graphical tools, which facilitated the comprehensive analysis of the experimental data. No additional statistical programs were used.

  • Lines 179-180; 189, 210; 235: Remove the bold

As suggested by the reviewer, the authors have removed the bold.

  • Line 182: The abbreviation is not necessary if you don´t use it more times throughout the manuscript.

In accordance with the suggestion of the reviewer, we have removed the abbreviation "PB" for phosphate buffer. Please see line 182.

  • Line 254: Results? Please check the IJMS format (Can be “results and discussion”)

As suggested by the reviewer, we have revised the section title to follow the IJMS format, changing “results” to "Results and Discussion”.

  • Line 266: Figure 1. Please separate each graph with letters “a” and “b” so that the reading of the results is clearer, e.g. Figure 1a and Figure 1b.

As reported by the reviewer, we have now separated the graphs in Figure 1 and labeled them as Figure 1a and Figure 1b for improved clarity. Additionally, we updated the figure caption to clearly describe each graph.

  • Lines 268-269: Placed the complete sentence at the end of the paragraph, after "extraction time..."

 As suggested by the reviewer, we have revised the text as follow “Statistical analysis confirmed no significant changes in yield with longer extraction times. Thus, to optimize energy consumption and minimize potential undesirable reactions, 45 minutes was chosen as the optimal extraction time. The total phenolic compounds did not significantly increase after 45 minutes of extraction (Figure 1a).”

Please see line 269-272

  • Lines 272-274: Please improve the writing. Do not repeat “Figure 1”so many times.

As highlighted by the reviewer" the authors have removed the phrase “As can be seen in Figure 1” and experiment number 8, Respectively.

  • Lines 277-282: Improve the writing. Why do you refer to Table 2 several times?

As indicated, we have revised the paragraph to eliminate the repeated references to "Table 2" and to enhance clarity. The updated version now consolidates all references into one, making the paragraph more concise and easier to follow. The revised text reads as follows: "Table 2 summarizes the extraction yields obtained under identical conditions (50 °C, 100 bar, 4 mL/min for 2 hours) using all the different solvent compositions from the mixture design. For comparison, SFE with pure COâ‚‚ at 350 bar, 50 °C, and 4 mL/min for 2 hours was also performed, and the results are included. Additionally, conventional maceration extractions were conducted using 100% acetone, 100% ethanol, and 100% water (see the Materials and Methods section for details).”

  • Lines 290-291: The information is repeated in the methodology. Focus on the results obtained.

As indicated by the reviewer, the authors have removed the phrase” Extraction yields for maceration with acetone, ethanol, and water were 10%, 15.4%, and 18%, respectively” to eliminate the repetitive information in the methodology section.

  • Lines 293-294: Please write the results obtained and move the table at the end of section 3.2

As requested by the reviewer, the authors have revised the text to present the results more clearly and have moved Table 2 to the end of section 3.2.

  • Line 295: It´s Table 2, not Table 1.

We appreciate the reviewer’s attention. The reference to Table 1 has been corrected to Table 2 to ensure accuracy.

  • Line 345: Why the statistical correlation not shown in the text, tables or figures?

Since it was simple linear correlation, we thought it were not interesting for readers, nevertheless, following reviewer suggestion we have included it in text TFC = 0.4329 TPC - 1.6715, R2 = 0.9442

  • Line 374: Remove “oxygen radical absorbance capacity”

As recommended by the reviewer, the authors have removed oxygen radical absorbance capacity.

Please see the line 369 page 7

  • Line 471: Although the analysis and interpretation of the results are correct, why were bibliographic references not included to strengthen these findings? The absence of citations in the discussion may be questionable. Please check this section.

Thanks for the observation. This section was devoted purely to show the results without discussion, we have included some references.  

  • Kühn, S.; Temelli, F. Recovery of Bioactive Compounds from Cranberry Pomace Using Ternary Mixtures of CO2 + Ethanol + Water. J. Supercrit. Fluids 2017, 130, 147–155, doi:10.1016/j.supflu.2017.07.028.
  • Radzali, S.A.; Markom, M.; Saleh, N.M. Co-Solvent Selection for Supercritical Fluid Extraction (SFE) of Phenolic Compounds from Labisia Pumila. Molecules 2020, 25, 1–15, doi:10.3390/molecules25245859.
  • Hassim, N.; Markom, M.; Rosli, M.I.; Harun, S. Scale-up Approach for Supercritical Fluid Extraction with Ethanol–Water Modified Carbon Dioxide on Phyllanthus Niruri for Safe Enriched Herbal Extracts. Sci. Rep. 2021, 11, 1–19, doi:10.1038/s41598-021-95222-0.
  • Solana, M.; Boschiero, I.; Dall’Acqua, S.; Bertucco, A. Extraction of Bioactive Enriched Fractions from Eruca Sativa Leaves by Supercritical CO2 Technology Using Different Co-Solvents. J. Supercrit. Fluids 2014, 94, 245–251, doi:10.1016/j.supflu.2014.08.022.
  • Lines 546-548: If you include details in the legend of table 4, why repeat them here? Simply refer to Table 4.

Following the reviewer's recommendation, the authors have replaced" the phrase “Table 4 provides specific details such as chromatographic retention time, peak area, classification into chemical families, compound name, chemical formula, molecular ion, and major ion fragments for each peak.” by “as shown in Table 4”.

  • Table 4. Remove “#” and add “pick”

As suggested by the reviewer, the authors have changed “#” and by “pick”. Please see table 4.

  • Lines 592; 658: Use a synonym instead of “in conclusion”. There is an exclusive section for conclusions.

As the reviewer indicated, the authors have replaced “in conclusion” by “To summarize “

  • Table 5: Remove “#”. Includes the meaning of “ND” at the end of the table.

As indicated by the reviewer, the authors have removed “#” and added the meaning of “ND”. Please the end of table 5.

  • Line 666: Please focus the conclusions in a clear and orderly manner on the fundamental aspects of the work: e.g. 1) Optimization of a pressurized extraction process based on a ternary solvent system. 2) Recovery of neuroprotective compounds from Eucalyptus marginata leaves. 3) In vitro assays.

Conclusion have been rewritten according to reviewer suggestion.

  • Line 684: Check the phrase “…and materials science”?

In keeping with the reviewer's observations, the sentence have been removed.

Reviewer 4 Report

Comments and Suggestions for Authors

In the article "Optimization of a pressurized extraction process based on a ternary solvent system for the recovery of neuroprotective compounds from Eucalyptus marginata leaves", the authors have explored sustainable extraction techniques to recover bioactive compounds from the leaves of Eucalyptus marginata plant. By combining supercritical fluid extraction (SFE), pressurized liquid extraction (PLE), and gas expanded liquid (GXL) in a ternary solvent system (CO2, ethanol, and water), the authors have aimed to explore the extraction efficiency. The extracted compounds were then evaluated for their neuroprotective activity, specifically their ability to inhibit acetylcholinesterase. Additionally, the study has identified 49 metabolites in the plant extracts using GC-MS and HPLC-ESI-MS/MS techniques. While the work is commendable, minor revisions could further emphasize the significance of the findings presented in this work. In this regard, the authors should address the following comments in their revised manuscript:

1.     The in-text citations and references (yellow highlights in the manuscript) should be formatted uniformly according to the journal's guidelines. Please ensure that reference numbers in the main text are not bolded.

2.     Similarly, headings like "Table and Figure" mentioned in the main text should not be bolded. Please make sure to follow the journal's guidelines.

3.     In section 2.1, Plant Material, how were the plants identified? The authors should provide the full accepted name of their plant under study. For this purpose, please visit the sites below and write the proper accepted full name of the plant. The authors should at least write the full accepted name of the plant in the Abstract section and in the experimental section.

https://www.ipni.org/   or

https://powo.science.kew.org/

4.     Line 236 and 237, please provide the version of the NIST and Wiley library used for the GC-MS studies in the revise manuscript.

5.     Line 250, was the LC-MS conducted only in negative ion mode?

Author Response

According to Reviewer: 4

  • The in-text citations and references (yellow highlights in the manuscript) should be formatted uniformly according to the journal’s guidelines. Please ensure that reference numbers in the main text are not bolded.

As suggested by the reviewer, the authors have revised the manuscript to ensure that all in-text citations and references are formatted uniformly in accordance with the journal’s guidelines. Specifically, we have removed the bold formatting from the reference numbers in the main text and ensured consistency throughout the manuscript.

  • Similarly, headings like “Table and Figure”; mentioned in the main text should not be bolded. Please make sure to follow the journal’s guidelines.

As reported by the reviewer, the authors have revised the manuscript to ensure that all headings such as "Table" and "Figure" in the main text are no bolded.

  1. In section 2.1, Plant Material, how were the plants identified? The authors should provide the full accepted name of their plant under study. For this purpose, please visit the sites below and write the proper accepted full name of the plant. The authors should at least write the full accepted name of the plant in the Abstract section and in the experimental section. https://www.ipni.org / or https://powo.science.kew.org/.

As recommended by the reviewer, the authors have provided the full scientific name of the plant and explained how it was identified by the botanist Dr. Hanene Ghazghazi in Tunisia. The full scientific name has been confirmed and is accepted according to authoritative databases such as Plants of the World Online (POWO) and the International Plant Names Index (IPNI). This name was first published in Transactions of the Linnean Society of London in 1802 and is classified under the Myrtaceae family. Eucalyptus marginata is native to Southwest Australia and primarily grows in subtropical biomes.

  1. Line 236 and 237, please provide the version of the NIST and Wiley library used for the GC-MS studies in the revise manuscript.

The following text has been included “Gas chromatography-mass spectrometry (GC-MS) analysis was conducted, and compound identification was achieved by comparing acquired mass spectra with reference spectra from the NIST Mass Spectral Library (version 09) and the Wiley Registry of Mass Spectral Data (8th edition).”

  1. 5.     Line 250, was the LC-MS conducted only in negative ion mode?

In light of the reviewer's comments. “The method used the negative ion mode for the mass spectrometer, which is often optimal for phenolic compounds, as they tend to ionize better in this mode. Negative ion mode was chosen in this case due to its effectiveness in detecting phenolic compounds, which are often better ionized under these conditions.”

Round 2

Reviewer 2 Report

Comments and Suggestions for Authors

Dear Authors,

The authors have revised or corrected the manuscript according to my mentioned comments and suggestions. I recommend it to be accepted for publication.

Reviewer 3 Report

Comments and Suggestions for Authors

The authors responded to comments and suggestions with clarity and scientificity.